# Intensive versus conservative glycemic control in patients undergoing coronary artery bypass graft surgery: A protocol for systematic review of randomised controlled trials

**Yi Liu, Xia-xuan Sun, Wen-ya Du, Ting-ting Chen, Meng Lv**◉*

Department of Anesthesiology, The First Affiliated Hospital of Shandong First Medical University & Shandong Provincial Qianfoshan Hospital, Ji'nan, Shandong Province, China

\* qylvmeng@163.com

## Abstract

### Introduction

Hyperglycemia and hypoglycemia are common during coronary artery bypass graft (CABG) and are associated with a variety of postoperative outcomes. Therefore, the strategy of intraoperative glycemic control is an important issue for the patients undergoing CABG. This systematic review aims to evaluate the effect of different intraoperative glycemic control strategies on postoperative outcomes.

### Methods and analyses

We will perform this systematic review of randomised controlled trials (RCTs) according to the recommendations of the Preferred Reporting Items for Systematic Reviews and Meta-Analyses (PRISMA). Relevant studies will be searched in Medline, Embase, Cochrane Library and Web of Science. Two independent reviewers will conduct study selection, data extraction, risk of bias and quality assessment. The primary outcome is postoperative mortality, and the secondary outcomes include the duration of mechanical ventilation in the intensive care unit (ICU), the incidence of postoperative myocardial infarction (MI), the incidence of postoperative atrial fibrillation (AF), the type and volume of blood product transfusion, the rate of rehospitalization, the rate of cerebrovascular accident, the rate of significant postoperative bleeding, the rate of infection, the incidence of acute kidney failure (AKF), hospital and ICU lengths of stay (LOS). ReviewManager 5.4 will be used for data management and statistical analysis. The Cochrane risk-of -bias tool 2.0 and GRADEpro will be applied for risk of bias and quality assessment of the evidence.

### Discussion

There is no consensus that which strategy of glycemic control is better for improving postoperative complications of patients undergoing CABG. The results of our study might provide some evidence for the relationship between intraoperative glycemic control strategies and postoperative outcomes in patients undergoing CABG.

**Data Availability Statement:** No datasets were generated or analysed during the current study. All relevant data from this study will be made available upon study completion.

**Funding:** Yes. Natural Science Foundation of Shandong Province (ZR2016HL02) play roles in study design and preparation of the manuscript.

**Competing interests:** The authors have declared that no competing interests exist.

# 1. Introduction

Nowadays, coronary artery bypass graft (CABG) surgery has become the most common surgical procedure. Almost 400,000 patients accept CABG annually [1]. Previous studies have shown that hyperglycemia during cardiac surgery is associated with hospital complications including mortality, renal failure, wound infections and the duration of mechanical ventilation, if it is not well controlled [2–6]. Besides, hyperglycemia has also been thought to increase perioperative morbidity and mortality [7].

On the other hand, it has been suggested that there may be a significant reduction in early mortality in patients who accepted the intensive glycemic control strategy during CABG [8–11]. However, other evidence of severe hypoglycemia resulted from intensive glycemic control brings the safety and effectiveness of intraoperative glycemic control strategy into question [5, 7, 12]. Moreover, hypoglycemia has also been considered as an independent risk factor of undesired clinical outcomes and hospital mortality [13]. Several studies confirmed that the incidence of hypoglycemia was associated with the overall risk of hospital mortality and the increased risk of cardiovascular events in critically ill patients [5, 14–16].

However, the optimal glycemic control strategy for patients undergoing CABG surgery remains controversial. Therefore, we will perform this systematic review and meta-analysis of randomised controlled trials to investigate the different effects of conservative and intensive glycemic control strategies on postoperative short-term mortality and severe complications in patients undergoing CABG.

# 2. Methods and analyses

## 2.1 Protocol design and registration

Our study has been registered at PROSPERO international prospective register of systematic reviews (https://www.crd.york.ac.uk/PROSPERO/). The registration number is CRD42021240841. The protocol is performed according to the Preferred Reporting Items for Systematic Reviews and Meta-Analyses Protocols (PRISMA-P) 2015 checklist [17]. The PRISMA-P 2015 checklist will be shown in S1 Checklist. We will conduct our systematic review following the Preferred Reporting Items for Systematic Reviews and Meta-analyses (PRISMA) 2020 statement [18].

## 2.2 Inclusion and exclusion criteria

**2.2.1 Study design.** Randomised controlled trials (RCTs) will be strictly screened. Crossover studies and quasi-randomised controlled trials will be excluded.

**2.2.2 Population.** Studies including patients undergoing CABG surgery who have accepted intraoperative glycemic control will be included. Studies including patients with incomplete information will be excluded.

**2.2.3 Intervention and comparator groups.** Studies with at least two glycemic control groups (intensive group and conservative group) will be included. In the systematic review, the group with a lower blood glucose target (≤160mg/dL) is defined as the 'intensive group', and the group with a higher blood glucose target (≤200mg/dL) is defined as the 'conservative group'. Studies that did not measure blood glucose or did not have both groups above will be excluded.

**2.2.4 Outcomes.** Studies that evaluate any variable of the primary or secondary outcomes as endpoints will be included. The primary outcome is postoperative mortality. The secondary outcomes include the variables as follow: the duration of mechanical ventilation in the intensive care unit (ICU), the incidence of postoperative myocardial infarction (MI), the incidence

of postoperative atrial fibrillation (AF), the type and volume of blood product transfusion, the rate of rehospitalization, the rate of cerebrovascular accident, the rate of significant postoperative bleeding, the rate of postoperative infection, the incidence of acute kidney failure (AKF), hospital and ICU lengths of stay (LOS). If the outcomes were reported on multiple time points, we will use the longest follow-up data.

All outcomes of our study are defined as follows:

1. Postoperative mortality: 30 days mortality after surgery or hospital mortality;

2. The duration of mechanical ventilation in ICU: from the time of admission to ICU to the time of tracheal extubation;

3. Postoperative MI: recognized by clinical features, including electrocardiographic (ECG) findings, elevated values of biochemical markers (biomarkers) of myocardial necrosis, and by imaging, or may be defined by pathology [19];

4. Postoperative AF: diagnosed with an electrocardiogram [20];

5. Blood product transfusion: the type and volume of blood products during the perioperative period;

6. Rehospitalization: re-admission due to postoperative complications of CABG;

7. Cerebrovascular accident: cerebrovascular accident within postoperative 30-day or during hospitalization;

8. Significant postoperative bleeding: postoperative bleeding ≥200 ml/h or ≥1000 ml in 24 hours, or massive bleeding needing reoperation;

9. Postoperative infection: any reported infection after surgery, such as pneumonia or wound infection;

10. AKF: new requirement for dialysis or an increase in serum creatinine > 2.0 mg/dl and double the most recent preoperative creatinine level;

11. ICU Length of stay (LOS): from ICU admission to transfer to the ward;

12. Hospital Length of stay (LOS): from hospital admission to the day of discharge.

**2.2.5 Language.** Studies reported in all types of languages will be included.

## 2.3 Search strategy

Two independent reviewers (YL and XXS) will perform the comprehensive search strategies in Medline, Embase, Cochrane Library and Web of Science. Records from the inception until the 9 August 2021 will be included. The search strategy will combine relevant Medical Subject Headings (MeSH) and keywords with synonyms and names of generic and brand names of the CABG and glycemic control. We will also check the reference lists of included RCTs and relevant reviews. In addition, a grey literature search will be performed on trial registry sites (such as WHO international clinical trial registry platform and NIH resources), regulatory agency databases for unpublished and ongoing studies. A manual search of citations and meeting reports will also be conducted. The details of search strategies of four databases were included in S1 File.

### 2.4 Study selection

Endnote X9 will be applied to manage relevant articles and remove duplicated. Two reviewers (YL and XXS) will assess the eligibility of the potentially relevant studies independently and in duplicate. Studies will be selected according to the predefined inclusion and exclusion criteria. The process will be conducted strictly in accordance with a study eligibility form (see S2 File). The detailed reasons for exclusion will be carefully documented. Any disagreement will be resolved by discussion or consulting a third reviewer (ML). If necessary, they will consult the methodological experts to reach consensus. The flow chart of study selection will be presented in S3 File.

### 2.5 Data extraction

The data will be extracted from eligible studies. A predefined data collection template will be used to collect the study data and will be tested on a few randomly selected studies. The data extraction template can be found in S1 Table. The author of study without insufficient information will be contacted for further information. Two investigators will perform data extraction independently and in duplicate. A third reviewer will be consulted to reach a consensus. Arm-level data (e.g., number of events) will be extracted whenever possible.

### 2.6 Risk of bias assessment

According to the Cochrane risk-of-bias tool 2.0 for the methodological quality of randomised trials [21], two investigators will independently perform the risk of bias of the included studies. The studies will be evaluated into three categories: low of risk, high of risk and some concerns. Bias of the following domains will be assessed: random sequence generation (selection bias), allocation concealment (selection bias), blinding of participants and personnel (performance bias), incomplete outcome data (attrition bias), selective reporting (reporting bias). In addition, we will reach an overall risk-of-bias judgement for the outcome based on the above assessments.

### 2.7 Measures of treatment effects

The level of blood glucose will be regarded as a binary variable. Each postoperative outcome will be estimated for intensive glycemic control group and conservative glycemic control group.

### 2.8 Assessment of heterogeneity

$\chi 2$ (Cochran's Q) and $I^2$ methods will be used to assess statistical heterogeneity among the eligible studies. According to the Cochrane Handbook for Systematic Reviews of Interventions, the $I^2$ method will be used to category the heterogeneity into non-important ($<$30%), moderate (30%-60%) and substantial ($>$60%).

### 2.9 Data synthesis

Risk ratio (RR) will be used for both fixed and random effects models (weighting by inverse of variance). The primary outcome and secondary outcomes of dichotomous variables will be assessed using forest plots and presented as RR. Sensitivity analyses will be performed to confirm the robustness of the results in the meta-analyses by excluding each study. A funnel plots for meta-analyses will be used to assess publication bias. If required, the Begg's and Egger's tests will be used to quantify the publication bias. Statistical analyses will be performed using RevMan5.4 software (the Nordic Cochrane Centre, Copenhagen, Denmark). For continuous

outcomes, weighted mean differences or standardized mean differences will be used according to different measurement scales. In our studies, the confidence interval (CI) will be established at 95%, and P value <0.05 will be considered to be statistically significant.

## 2.10 Subgroup analyses

To explore the sources of the heterogeneity of the included studies, subgroup analysis will be performed according to the primary and secondary outcomes. We will conduct subgroup analyses based on the following criteria:

1. with versus without the history of diabetes;

2. history of diabetes ≤5 years versus >5 years;

3. with versus without the history of hypertension;

4. with versus without the history of smoking;

5. different glycemic control strategy: insulin versus GIK;

6. off-pump CABG versus on-pump CABG.

## 2.11 Sensitivity analyses

To confirm the robustness of our findings, a sensitivity analysis will be conducted based on the different levels of bias of the included studies. To evaluate the internal validity of studies or treatment adequacy, we will subsequently remove studies of 'high risk of bias', studies of 'some concerns', and studies of 'low risk of bias'. In addition, both random effect model and fixed effect model will be used for the synthesis of the primary and secondary outcomes.

## 2.12 Meta-bias

To determine whether reporting bias is present, we will explore whether the protocol of the studies included in this systematic review was published before recruitment of patients was started. The reporting bias will be further determined by funnel plots if there are more than 10 studies in this systematic review.

## 2.13 Confidence in cumulative estimate

In order to evaluate the quality of the treatment effect estimate, the Grading of Recommendations Assessment, Development and Evaluation (GRADE) approach will be used. We will rate the quality of evidence by using the five GRADE categories: risk of bias, inconsistency, indirectness, imprecision and publication bias. In addition, we will categorize confidence in the estimate of the outcomes into four levels: high level, moderate level, low level and very low level [22]. The GRADEpro software will be applied to estimate the quality of evidence assessment (see S4 File).

## 2.14 Patient and public involvement

There will be no patients and/or the public involved in the design, or conduct, or reporting, or dissemination plans of this study.

## 3. Discussion

Intraoperative blood glucose level is considered as a significant factor of postoperative outcomes. However, there is no consensus that which strategy of glycemic control is more beneficial for improving postoperative complications of patients undergoing CABG. The aim of this systematic review is to explore the effects of different glycemic control strategies on postoperative outcomes of patients undergoing CABG.

In our study, we will conduct a comprehensive database search, select eligible studies, extract data, assess the risk of bias, perform data synthesis and assess the heterogeneity among included studies. Moreover, subgroup analysis will be conducted to explore the source of heterogeneity, and sensitivity analysis will be performed to ascertain the robustness of the results. All of above process will be achieved by two reviewers independently and in duplicate.

There are two limitations in our study. First, some perioperative complications of CABG only reported in few studies, which limits the meta-analysis for these complications. Second, some of the eligible studies included in our systematic review might be low quality. We will perform sensitivity analysis to ascertain the robustness of the results.

## Supporting information

**S1 Checklist. PRISMA-P 2015 checklist.**
(DOCX)

**S1 Table. Draft of data extraction template.**
(DOCX)

**S1 File. Search strategies.**
(DOCX)

**S2 File. Study eligibility form.**
(DOCX)

**S3 File. PRISMA 2020 flow diagram.**
(TIF)

**S4 File. Assessment of evidence quality by GRADE approach.**
(DOCX)

## Acknowledgments

The authors would like to thank the First Affiliated Hospital of Shandong First Medical University for their assistance and support in conducting this systematic review.

## Author Contributions

**Conceptualization:** Yi Liu, Meng Lv.

**Data curation:** Yi Liu, Xia-xuan Sun, Wen-ya Du.

**Formal analysis:** Xia-xuan Sun, Wen-ya Du, Ting-ting Chen.

**Project administration:** Yi Liu, Ting-ting Chen, Meng Lv.

**Supervision:** Meng Lv.

**Writing – original draft:** Yi Liu, Xia-xuan Sun.

**Writing – review & editing:** Meng Lv.

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
