## [Decision Letter · Decision Letter 0]

19 Jul 2022

PONE-D-22-18985Intensive versus conservative glycaemic control in patients undergoing coronary artery bypass graft surgery: A protocol for systematic review of randomised controlled trials PLOS ONE

Dear Dr. Lv,

Thank you for submitting your manuscript to PLOS ONE. After careful consideration, we feel that it has merit but does not fully meet PLOS ONE’s publication criteria as it currently stands. Therefore, we invite you to submit a revised version of the manuscript that addresses the points raised during the review process.

We look forward to receiving your revised manuscript.

Kind regards,

Ozra Tabatabaei-Malazy

Academic Editor

PLOS ONE

Journal Requirements:

Additional Editor Comments:

Reviewer#1:

The authors have proposed a protocol for a systematic review on intensive vs. conservative glycemic control in CABG patients. The methods have been well-described, and I hope the authors stick to all these elements, particularly critical appraisal and bias control. Although the review protocol has been already registered in PROSPERO, I suggest these comments be considered in the review:

1- Include the duration of the disease as a confounder in your meta-analysis.

2- I think you can add more variables to the secondary outcomes, such as the need for blood product transfusion, postoperative atrial fibrillation, and rehospitalization.

3- Subanalysis section: Please consider a better wording for "not accompanied of." That does not sound nice!

4- There are some typos and grammatical issues in the manuscript. So please have it checked by an expert.

Reviewer#2:

There are several points in this protocol that require further clarification, as follows:

Introduction:

- The authors clearly have described the condition, how glycemic control during cardiac surgery might affect postoperative outcomes, and how different glycemic control strategies might work.

- The purpose of the study has been explained in a very general way. The authors should specify exactly what research questions this review ultimately answers.

- Place the point after the closing bracket of the citations.

Methods:

- The authors should describe some items of eligibility criteria more clearly and in detail:

1) Regarding the target population, will you include all patients undergoing CABG surgery and intraoperative glycemic control, regardless of their demographic characteristics (e.g., age), background, and past medical history of diabetes mellitus?

2) Regarding types of interventions, please explain clearly what range of blood sugar control you consider to define “intensive” and “conservative” control.

3) Please specify the comparison group more clearly. Will you only include studies that considered high levels of glycemic control target (conservative group) as the comparison group?

4) Since you discussed the eligibility criteria based on the PICOS in the 2.2 subsection, and the outcomes are also one of its items, there is no need to describe the outcomes in a separate section. Please leave the explanation of subsection 2.3, which is about the outcomes, below in subsection 2.2.

5) The authors should describe the report characteristics of studies to be included, such as years considered, language, and publication status. If relevant studies published in all languages are to be included, how to deal with articles that have a language different from the language of the authors?

- Study selection, data synthesis, and risk of bias assessment have been described according to the PRISMA-P statement.

Other comments:

- The authors should arrange the order of the abstract sections according to the main text (Introduction/background, Methods, Discussion).

- The manuscript has some typing mistakes and grammatical errors that need to be corrected

Reviewers' comments:

Reviewer's Responses to Questions

**Comments to the Author**

1. Does the manuscript provide a valid rationale for the proposed study, with clearly identified and justified research questions?

Reviewer #1: Yes

Reviewer #2: Partly

2. Is the protocol technically sound and planned in a manner that will lead to a meaningful outcome and allow testing the stated hypotheses?

Reviewer #1: Yes

Reviewer #2: Partly

3. Is the methodology feasible and described in sufficient detail to allow the work to be replicable?

Reviewer #1: Yes

Reviewer #2: Yes

4. Have the authors described where all data underlying the findings will be made available when the study is complete?

Reviewer #1: Yes

Reviewer #2: Yes

5. Is the manuscript presented in an intelligible fashion and written in standard English?

Reviewer #1: Yes

Reviewer #2: Yes

6. Review Comments to the Author

You may also provide optional suggestions and comments to authors that they might find helpful in planning their study.

Reviewer #1: The authors have proposed a protocol for a systematic review on intensive vs. conservative glycemic control in CABG patients. The methods have been well-described, and I hope the authors stick to all these elements, particularly critical appraisal and bias control. Although the review protocol has been already registered in PROSPERO, I suggest these comments be considered in the review:

1- Include the duration of the disease as a confounder in your meta-analysis.

2- I think you can add more variables to the secondary outcomes, such as the need for blood product transfusion, postoperative atrial fibrillation, and rehospitalization.

3- Subanalysis section: Please consider a better wording for "not accompanied of." That does not sound nice!

4- There are some typos and grammatical issues in the manuscript. So please have it checked by an expert.

Reviewer #2: In this protocol, the authors tried to describe the objectives, eligibility criteria, and planned methodology of their systematic review that will assess the effect of different intraoperative glycemic control strategies on postoperative outcomes. Evidence shows that poor intraoperative glycemic control is a risk factor for postoperative morbidity and mortality. Different strategies are used for intraoperative control of blood sugar, but their effect on postoperative outcomes has been conflicting across studies. Therefore, this systematic review aims to examine this crucial question and, more specifically, to determine the effect intensive glycemic control of blood sugar has on postoperative outcomes versus conservative control. The title is interesting and the protocol has been written according to the PRISMA-P statement. However, there are several points in this protocol that require further clarification, as follows:

Introduction:

- The authors clearly have described the condition, how glycemic control during cardiac surgery might affect postoperative outcomes, and how different glycemic control strategies might work.

- The purpose of the study has been explained in a very general way. The authors should specify exactly what research questions this review ultimately answers.

- Place the point after the closing bracket of the citations.

Methods:

- The authors should describe some items of eligibility criteria more clearly and in detail:

1) Regarding the target population, will you include all patients undergoing CABG surgery and intraoperative glycemic control, regardless of their demographic characteristics (e.g., age), background, and past medical history of diabetes mellitus?

2) Regarding types of interventions, please explain clearly what range of blood sugar control you consider to define “intensive” and “conservative” control.

3) Please specify the comparison group more clearly. Will you only include studies that considered high levels of glycemic control target (conservative group) as the comparison group?

4) Since you discussed the eligibility criteria based on the PICOS in the 2.2 subsection, and the outcomes are also one of its items, there is no need to describe the outcomes in a separate section. Please leave the explanation of subsection 2.3, which is about the outcomes, below in subsection 2.2.

5) The authors should describe the report characteristics of studies to be included, such as years considered, language, and publication status. If relevant studies published in all languages are to be included, how to deal with articles that have a language different from the language of the authors?

- Study selection, data synthesis, and risk of bias assessment have been described according to the PRISMA-P statement.

Other comments:

- The authors should arrange the order of the abstract sections according to the main text (Introduction/background, Methods, Discussion).

- The manuscript has some typing mistakes and grammatical errors that need to be corrected

7. PLOS authors have the option to publish the peer review history of their article (what does this mean?). If published, this will include your full peer review and any attached files.

Reviewer #1: **Yes: **Akbar Shafiee

Reviewer #2: **Yes: **Danesh Soltani

---

## [Author Response · Author response to Decision Letter 0]

30 Aug 2022

Dear editor and reviewers,

Thanks for your letter and the reviewers’ comments concerning our manuscript entitled “Intensive versus conservative glycemic control in patients undergoing coronary artery bypass graft surgery: A protocol for systematic review of randomized controlled trials” (PONE-D-22-18985). The comments are very helpful and valuable for revising and improving our paper. We would like to thank you for allowing us to submit a revised copy of the manuscript, and we highly appreciate your time and consideration. We have read through those comments carefully and revised the manuscript accordingly. In addition to the response letter, a marked and clean version of our revised manuscript were uploaded based on the instructions provided in your letter. Revised portions are marked in red in the marked version of revised manuscript. Our point-by-point responses are detailed below.

Reviewer#1:

Q1. Include the duration of the disease as a confounder in your meta-analysis.

Response: We deeply appreciate the suggestion. According to the reviewer’s comment, we will include the duration of some diseases (e.g., the duration of diabetes mellitus, coronary heart disease and hypertension) as confounders. The characteristics of all patients in eligible literatures will be carefully extracted and recorded in detail for subsequent analyses. All items that need to be extracted were presented in the predefined table (see S1 table. Draft of data extraction template). 

Q2. I think you can add more variables to the secondary outcomes, such as the need for blood product transfusion, postoperative atrial fibrillation, and rehospitalization.

Response: Thank you for your precious advice. It is of great interest to add more variables to the secondary outcomes. Therefore, we decide to add the type and volume of blood product transfusion, postoperative atrial fibrillation, rehospitalization, cerebrovascular accident, and significant postoperative bleeding as secondary outcomes. (Line 114, page 6)

Q3. Subanalysis section: Please consider a better wording for "not accompanied of." That does not sound nice!

Response: Thank you for your thoughtful suggestions. In the revised manuscript, we have replaced “not accompanied of” with the word “without”, and we have modified this expression throughout the text according to the reviewer’s comment. (Line 205, 207 and 208, page 11) 

Q4. There are some typos and grammatical issues in the manuscript. So please have it checked by an expert.

Response: Thank you for your careful review. We have corrected the typos and grammatical issues. The revision portions were marked in red in the revised manuscript. In order to meet the standard of the journal, the manuscript has been checked by a professional polishing company named American journal expert (AJE, verification code 67E6-E5CE-98E4-68EB-5D9A). 

Reviewer#2:

The part of introduction:

Q1. The authors clearly have described the condition, how glycemic control during cardiac surgery might affect postoperative outcomes, and how different glycemic control strategies might work. 

Response: Thanks very much for taking your time to review our manuscript. The reviewer’s comments were highly insightful and enabled us to greatly improve the quality of our manuscript. 

Q2. The purpose of the study has been explained in a very general way. The authors should specify exactly what research questions this review ultimately answers. 

Response: Thank you very much for your precious suggestions. We will perform this systematic review and meta-analysis to investigate the effect of different glycemic control strategies on the postoperative outcomes in patients undergoing coronary artery bypass graft. To be more exact and in accordance with the reviewer’ comment, we have clearly demonstrated the purpose of this review in the revised manuscript. (Line 78-82, page 4-5)

Q3. Place the point after the closing bracket of the citations.

Response: We deeply appreciate the reviewer’s suggestions. According to the comments, we have placed the point after the citations and have carefully checked the entire manuscript.

The part of methods:

Q1. Regarding the target population, will you include all patients undergoing CABG surgery and intraoperative glycemic control, regardless of their demographic characteristics (e.g., age), background, and past medical history of diabetes mellitus?

Response: Thank you for the thoughtful comments. All patients undergoing CABG surgery and intraoperative glycemic control will be included in our study, regardless of their demographic characteristics (e.g., age and sex), the history and duration of diabetes mellitus, the duration of coronary artery disease, the history and duration of hypertension, and the history of smoking. The items that needed to be extracted had been presented in the predefined table (see S1 table. Draft of data extraction template). All detailed information would be extracted according to the draft when the protocol is implemented. Moreover, we will perform secondary analyses (e.g., subgroup analysis and sensitivity analysis) according to demographic characteristics and past medical history. (Line 205-208, page 11)

Q2. Regarding types of interventions, please explain clearly what range of blood sugar control you consider to define “intensive” and “conservative” control.

Response: Thank you for the reviewer’s suggestions. In this systematic review, the participants of the included studies were allocated into different groups based on the target blood glucose level. The group with a lower target blood glucose level (≤160mg/dL) was defined as the intensive group, while the other group with a higher target blood glucose level (≤200mg/dL) was defined as the conservative group. (Line 104-108, page 6)

Q3. Please specify the comparison group more clearly. Will you only include studies that considered high levels of glycemic control target (conservative group) as the comparison group? 

Response: We deeply appreciate the reviewer’s careful review. All studies with at least two glycemic control group (intensive versus conservative group) will be included in our systematic review. (Line 104-105, page 6).

Q4. Since you discussed the eligibility criteria based on the PICOS in the 2.2 subsection, and the outcomes are also one of its items, there is no need to describe the outcomes in a separate section. Please leave the explanation of subsection 2.3, which is about the outcomes, below in subsection 2.2.

Response: Thank you very much for the reviewer’s valuable advice. We strongly agree with the comments and leave the explanation 2.3 below in subsection 2.2 in the revised manuscript (Line 111-140, page 6-7). 

Q5. The authors should describe the report characteristics of studies to be included, such as years considered, language, and publication status. If relevant studies published in all languages are to be included, how to deal with articles that have a language different from the language of the authors?

Response: Thank you for your careful review and we strongly agree with your comment. There is no limitation on publication year, journal, first author, language, and publication status. The above literatures characteristics have been taken into account (see S1 table). If some literatures with languages different from the authors’ language, we would seek help on some translation software (e.g., Youdao Translator, Google Translator and Baidu Translator). Moreover, we would consult experts proficient in languages.

Other comments:

Q1. The authors should arrange the order of the abstract sections according to the main text (Introduction/background, Methods, Discussion).

Response: We are extremely grateful to the reviewer for pointing out this problem. In accordance with the reviewer’s comments, we would like to arrange the order of the abstract sections according to the main text consisting of three parts (introduction, methods and analyses, discussion).

Q2. The manuscript has some typing mistakes and grammatical errors that need to be corrected

Response: Thank you for your valuable suggestions. Following the reviewer’s suggestions, we carefully checked and corrected the mistakes and grammatical errors one by one. And the corrections were marked in red in the marked version.

Our deepest gratitude goes to you for your careful work and thoughtful suggestions that would help to improve this paper substantially. As the reviewers’ suggestions, we tried our best to improve the manuscript and made some changes in the manuscript. We appreciate for the editors and reviewers’ warm work earnestly, and hope that the corrections will meet with approval.

Once again, thank you very much for your efforts in reviewing our manuscript.

Your sincerely,

Meng Lv

Department of Anesthesiology, The First Affiliated Hospital of Shandong First Medical University & Shandong Provincial Qianfoshan Hospital, Ji'nan, Shandong Province, 250014, China.

Telephone: +8615169105373

Email: qylvmeng@163.com

---

## [Decision Letter · Decision Letter 1]

12 Sep 2022

PONE-D-22-18985R1Intensive versus conservative glycaemic control in patients undergoing coronary artery bypass graft surgery: A protocol for systematic review of randomised controlled trialsPLOS ONE

Dear Dr. Lv,

Thank you for submitting your manuscript to PLOS ONE. After careful consideration, we feel that it has merit but does not fully meet PLOS ONE’s publication criteria as it currently stands. Therefore, we invite you to submit a revised version of the manuscript that addresses the points raised during the review process.

We look forward to receiving your revised manuscript.

Kind regards,

Ozra Tabatabaei-Malazy

Academic Editor

PLOS ONE

Journal Requirements:

Additional Editor Comments:

Reviewer#1:

The authors have well revised the manuscript; however, I can still see some errors in writing, particularly in using the articles (i.e., the). As an example from the abstract, it should be written: "the intensive care unit (ICU), the incidence of postoperative myocardial infarction), the incidence of ....". Or in discussion of the Abstract "There is no certain evidence that which strategy of glycemic...." is better to change "There is no certain evidence to clarify which strategy of glycemic....". Please double check the manuscript for these issues.

Reviewers' comments:

Reviewer's Responses to Questions

**Comments to the Author**

1. Does the manuscript provide a valid rationale for the proposed study, with clearly identified and justified research questions?

Reviewer #1: Yes

Reviewer #2: Yes

2. Is the protocol technically sound and planned in a manner that will lead to a meaningful outcome and allow testing the stated hypotheses?

Reviewer #1: Yes

Reviewer #2: Yes

3. Is the methodology feasible and described in sufficient detail to allow the work to be replicable?

Reviewer #1: Yes

Reviewer #2: Yes

4. Have the authors described where all data underlying the findings will be made available when the study is complete?

Reviewer #1: Yes

Reviewer #2: Yes

5. Is the manuscript presented in an intelligible fashion and written in standard English?

Reviewer #1: No

Reviewer #2: Yes

6. Review Comments to the Author

You may also provide optional suggestions and comments to authors that they might find helpful in planning their study.

Reviewer #1: The authors have well revised the manuscript; however, I can still see some errors in writing, particularly in using the articles (i.e., the). As an example from the abstract, it should be written: "the intensive care unit (ICU), the incidence of postoperative myocardial infarction), the incidence of ...." Please double check the manuscript for these issues.

Reviewer #2: The protocol now contains all of the necessary details to conduct a standard systematic review study. What I'd like to emphasize is that the authors strictly follow the protocol. I have nothing further to add.

7. PLOS authors have the option to publish the peer review history of their article (what does this mean?). If published, this will include your full peer review and any attached files.

Reviewer #1: No

Reviewer #2: **Yes: **Danesh Soltani

---

## [Author Response · Author response to Decision Letter 1]

27 Sep 2022

Dear editor and reviewers,

Thanks for your letter and the reviewers’ comments concerning our manuscript entitled “Intensive versus conservative glycemic control in patients undergoing coronary artery bypass graft surgery: A protocol for systematic review of randomised controlled trials” (PONE-D-22-18985). The comments are very helpful and valuable for revising and improving our manuscript. We would like to thank you for allowing us to submit a revised copy of the manuscript, and we highly appreciate your time and consideration. We have read through those comments carefully and revised the manuscript accordingly. In addition to the response letter, a marked and clean version of our revised manuscript were uploaded based on the instructions provided in your letter. Revised portions are marked in red in the marked version of revised manuscript.

Reviewer#1:

The authors have well revised the manuscript; however, I can still see some errors in writing, particularly in using the articles (i.e., the). As an example from the abstract, it should be written: "the intensive care unit (ICU), the incidence of postoperative myocardial infarction), the incidence of ....". Or in discussion of the Abstract "There is no certain evidence that which strategy of glycemic...." is better to change "There is no certain evidence to clarify which strategy of glycemic....". Please double check the manuscript for these issues.

Response: Thank you for your valuable suggestions. Following the reviewer’s suggestions, we carefully corrected the mistakes and double checked the manuscript. And the corrections were marked in red in the marked version.

Once again, thank you very much for your efforts in reviewing our manuscript.

Your sincerely,

Meng Lv

Department of Anesthesiology, The First Affiliated Hospital of Shandong First Medical University & Shandong Provincial Qianfoshan Hospital, Ji'nan, Shandong Province, 250014, China.

Telephone: +8615169105373

Email: qylvmeng@163.com

---

## [Editor Report · Decision Letter 2]

4 Oct 2022

Intensive versus conservative glycemic control in patients undergoing coronary artery bypass graft surgery: A protocol for systematic review of randomised controlled trials

PONE-D-22-18985R2

Dear Dr. Lv,

We’re pleased to inform you that your manuscript has been judged scientifically suitable for publication and will be formally accepted for publication once it meets all outstanding technical requirements.

Kind regards,

Ozra Tabatabaei-Malazy

Academic Editor

PLOS ONE

---

## [Editor Report · Acceptance letter]

7 Oct 2022

PONE-D-22-18985R2 

Intensive versus conservative glycemic control in patients undergoing coronary artery bypass graft surgery: A protocol for systematic review of Randomised controlled trials 

Dear Dr. Lv:

I'm pleased to inform you that your manuscript has been deemed suitable for publication in PLOS ONE. Congratulations! Your manuscript is now with our production department. 

Kind regards, 

on behalf of

Dr. Ozra Tabatabaei-Malazy 

Academic Editor

PLOS ONE